# Plant-Derived Substances for Prevention of Necrotising Enterocolitis: A Systematic Review of Animal Studies

**DOI:** 10.3390/nu16060832

**Published:** 2024-03-14

**Authors:** Cheryl Anne Mackay, Chandra Rath, Shripada Rao, Sanjay Patole

**Affiliations:** 1Neonatology, King Edward Memorial Hospita, Subiaco 6008, Australia; 2Perth Children’s Hospital, Nedlands 6009, Australia; 3School of Medicine, University of Western Australia, Crawley 6009, Australia

**Keywords:** necrotizing enterocolitis, plant-derived substance, animal, newborn

## Abstract

Inflammation, oxidative injury, and gut dysbiosis play an important role in the pathogenesis of necrotising enterocolitis (NEC). Plant-derived substances have historically been used as therapeutic agents due to their anti-inflammatory, antioxidant, and antimicrobial properties. We aimed to review pre-clinical evidence for plant-derived substances in the prevention and treatment of NEC. A systematic review was conducted using the following databases: PubMed, EMBASE, EMCARE, MEDLINE and Cochrane Library (PROSPERO CRD42022365477). Randomized controlled trials (RCTs) and quasi-RCTs that evaluated a plant-derived substance as an intervention for NEC in an animal model of the illness and compared pre-stated outcomes (e.g., clinical severity, severity of intestinal injury, mortality, laboratory markers of inflammation and oxidative injury) were included. Sixteen studies (*n* = 610) were included in the systematic review. Ten of the sixteen included RCTs (Preterm rat pups: 15, Mice: 1) reported mortality and all reported NEC-related histology. Meta-analysis showed decreased mortality [12/134 vs. 27/135; RR: 0.48 (95% CI: 0.26 to 0.87); *p* = 0.02, 10 RCTs] and decreased NEC in the experimental group [24/126 vs. 55/79; RR: 0.34 (95% CI: 0.22 to 0.52); *p* < 0.001, 6 RCTs]. Markers of inflammation (*n* = 11) and oxidative stress (*n* = 13) improved in all the studies that have reported this outcome. There was no significant publication bias for the outcome of mortality. Plant-derived substances have the potential to reduce the incidence and severity of histologically diagnosed NEC and mortality in rodent models. These findings are helpful in guiding further pre-clinical studies towards developing a food supplement for the prevention of NEC in preterm infants.

## 1. Introduction

Necrotising enterocolitis (NEC) is a potentially devastating illness, particularly in extremely preterm (gestation < 28 weeks) infants [1,2]. Overall, the incidence of NEC ≥ Stage II is reported to be 4 to 6% in very preterm infants, rising to 8 to 12% in extremely preterm infants with higher mortality (~20% vs. 45%) and morbidity, including long-term neurodevelopmental impairment [1,2,3,4,5]. The complications associated with NEC, such as prolonged hospital stay and short bowel syndrome, explain the enormous socioeconomic burden of the illness, reported to be as high as USD 500 million to 1 billion per year in the USA [6].

Despite decades of research, the pathogenesis of NEC continues to be poorly understood. An interplay between prematurity, hypoxia-ischemia-reperfusion-free radical injury, gut dysbiosis, and formula feeding is considered important in the pathogenesis [1,2]. A gestational age-dependent excessive pro-inflammatory response following interaction between Toll-like receptor-4 (TLR-4) and lipopolysaccharides (LPS) is currently considered an important trigger for gut injury in NEC in preterm infants involving oxidative stress [1,2,7,8,9,10,11]. Current strategies for the prevention of NEC include antenatal glucocorticoids, early preferential feeding with breast milk, standardised feeding protocols, probiotic supplementation, avoiding formula feeding and undue prolonged exposure to antibiotics, and acid-supressing agents [12,13,14].

The need for new strategies for prevention and treatment of NEC cannot be overemphasized considering the health burden associated with the illness. Plant-derived substances including herbs and spices may provide a novel option in this context based on their anti-inflammatory, antioxidant, and antimicrobial properties, and the fact that they have been used as therapeutic agents for centuries [15,16,17,18,19,20,21,22,23]. Importantly, pre-clinical studies have shown significant benefits of ginger, curcumin, and *Nigella sativa* oil in reducing the severity of intestinal injury and markers of oxidative injury in preterm rat pup models of NEC [24,25,26,27,28]. Given these data, we aimed to review the evidence on plant-derived substances as a potentially novel option for the prevention and treatment of NEC in preterm infants. Our findings are expected to guide translational research in this field.

## 2. Materials and Methods

We used the animal intervention study (SYRCLE) protocol and the preferred reporting items for systematic reviews and meta-analyses (PRISMA) guidelines for conducting and reporting this systematic review, respectively [29,30]. The protocol was registered on PROSPERO, the international prospective register of systematic reviews (CRD42022365477).

### 2.1. Data Sources and Searches

The databases PubMed, EMBASE (through OVID), EMCARE (through OVID), MEDLINE (through OVID), Cochrane Library, and Google Scholar were searched (since their inception until December 2022) independently by three reviewers. The ClinicalTrials.gov website was searched to identify ongoing clinical studies. Grey literature was searched through ‘Mednar’ (http://mednar.com/mednar/desktop/en/search.html, accessed on 20 December 2022) database. The reference lists of eligible studies and review articles were hand searched to identify additional studies. No language restrictions were applied. PubMed was searched using the following broad keywords: (Plant) AND (((Necrotising enterocolitis) OR (Necrotizing enterocolitis)) OR (NEC)). That automatic mapping system of PubMed expanded it to cover all the following terms: (“plant s”[All Fields] OR “planted”[All Fields] OR “planting”[All Fields] OR “plantings”[All Fields] OR “plants”[MeSH Terms] OR “plants”[All Fields] OR “plant”[All Fields]) AND (“necrotising enterocolitis”[All Fields] OR “enterocolitis, necrotizing”[MeSH Terms] OR (“enterocolitis”[All Fields] AND “necrotizing”[All Fields]) OR “necrotizing enterocolitis”[All Fields] OR (“necrotizing”[All Fields] AND “enterocolitis”[All Fields])).

### 2.2. Study Selection

Eligibility and exclusion criteria: Pre-clinical randomized controlled trials (RCTs) and quasi-RCTs meeting the following criteria were eligible for inclusion: (1) Use of a validated animal model of NEC in preterm infants. (2) Evaluation of a plant-derived substance as an experimental intervention for NEC. (3) Comparison of pre-stated clinical and laboratory outcomes between the experimental and control group. Observational and in vitro studies, and studies that used unvalidated models, were excluded.

Outcomes: These included the following: (1) Clinical assessment of the severity of NEC. (2) Severity of gut injury assessed by histological grading. (3) Mortality from NEC. (4) Laboratory markers of inflammation and oxidative injury. (5) Other outcomes.

### 2.3. Data Extraction and Quality Assessment

Two authors independently screened the titles and abstracts to identify studies potentially eligible for inclusion in the review. Full-text articles of such studies were read by two reviewers to confirm their eligibility for inclusion. A standardised piloted form was used to extract data. The incidence of various clinical outcomes of interest in the experimental vs. control group was recorded. We used risk ratios (RR) and 95% confidence intervals (CI) for dichotomous outcomes. All authors were contacted for additional information. The quality of the RCTs was assessed using the SYRCLE risk of bias (ROB) tool [29]. Two authors independently assessed the ROB in the domains of random number generation, allocation concealment, random housing of animals, blinding of intervention and outcome assessors, selective reporting, and other potential sources of bias. For each domain, the risk was assessed as low, high, or unclear The certainty of evidence (COE) was assessed using the GRADE methodology and classified into one of the four categories: high, moderate, low and very low [31]. In case of discrepancies, group discussions involving all reviewers were held to reach consensus.

### 2.4. Data Synthesis

We conducted meta-analysis using the Stata 16.0 software (StataCorp. 2019. Stata Statistical Software: Release 16. College Station, TX, USA: StataCorp LLC). We used the DerSimonian and Laird random-effects model (REM) for meta-analysis since heterogeneity was expected. We used raw numbers to calculate the RRs for pooling if the included studies did not provide this information. For dichotomous outcomes, the pooled effect estimates were presented as pooled RRs with 95% CIs. Qualitative synthesis and diagrammatic representation were provided for studies where meta-analysis was not possible. We assessed statistical heterogeneity by visual inspection of the forest plots and quantified this using the I^2^ statistic. The I^2^ results were interpreted as follows: 0% to 40%: heterogeneity might not be important; 30% to 60%: may represent moderate heterogeneity; 50% to 90%: may represent substantial heterogeneity; 75% to 100%: considerable heterogeneity [32]. We assessed publication bias (small study effects) using visual inspection of the funnel plots, Egger’s test, Begg’s test, and Harbord’s test if ≥10 studies were included for any individual outcome.

## 3. Results

A PRISMA flow chart of screening and selection results is shown in Figure 1. The initial search identified 544 articles of which 16 studies [25,27,28,33,34,35,36,37,38,39,40,41,42,43,44,45] (*n* = 610) were included after application of the selection criteria. Two studies were excluded because they evaluated a genetically engineered analogue of plant-derived substance [26,46]. All included studies were RCTs that evaluated a plant-derived substance in an animal model of NEC (Table 1). The ROB was generally high for the included studies. Majority of the included studies carried unclear or high ROB for randomization or reporting (Table 2).

(1)Animal model of NEC: Except for Fang et al. [38], whose work involved mice pups, all included studies were conducted in the preterm rat pup model of NEC. The median (IQR) sample size was 36 (10). NEC was induced by Hypoxia-reoxygenation and hypothermia (*n* = 8) [27,28,34,36,39,40,44,45], hypoxia-reoxygenation (*n* = 1) [37], hyperosmolar feeds, hypoxia-reoxygenation and hypothermia (*n* = 1) [35], LPS administration, hyperosmolar formula feed, hypoxia-reoxygenation and hypothermia (*n* = 1) [43], LPS administration, hypoxia-reoxygenation and hypothermia (*n* = 2) [25,33], LPS administration and hypoxia-reoxygenation (*n* = 2) [41,42], and acetic acid enema (*n* = 1) [38].(2)Age at euthanasia of animals: Rodents were euthanised on day 4 in eleven of the included studies [25,27,33,34,35,37,39,40,43,44,45], day 5 in three included studies [36,41,42], and day 10 in one included study [38]. One study did not provide this information [28].(3)Histopathological grading of intestinal injury: Following euthanasia the animal’s abdomen was dissected, and the intestinal samples stained with haematoxylin and eosin (H&E) were studied under magnification. A histological score was assigned from 0 to 4 where 0 = normal intestine; 1 = mild injury affecting villus tips, with or without separation of villus-core; 2 = moderate injury with extensive villus-core separation and loss of villi; 3 = extensive epithelial sloughing, loss of villi and/or necrosis extending to submucosal level; 4 = transmural necrosis or perforation.(4)Single vs. combination product: A single plant-derived substance was evaluated in 14 RCTs including sumac (*Rhus chinensis/Rhus coriaria*) [39,45], curcumin [28,40], fennel extract [43], sesamol [35], berberine [38,41], ginger [25], quercetin [44], nigella sativa oil [27], pomegranate seed oil [36], astragaloside [34], and resveratrol [37]. The remaining two RCTs assessed a combination of plant-derived substances including Ankaferd Blood Stopper (containing Thymus vulgaris, Glycyrrhiza glabra, Vitis vinifera, Alpinia officinarum and Urtica dioica) [33] and daikenchuto (containing ginger, ginseng and zanthoxylum fruit) [42].(5)Route of administration: Thirteen studies used the oral route of administration of the plant-derived substance [25,27,28,34,36,37,38,39,41,42,43,44,45], two studies used the intraperitoneal route [33,40], and another involved both the oral as well as intraperitoneal route [35].(6)Phytochemical levels: Four RCTs assessed the predominant phytochemicals in the plant-derived substance [25,39,43,45].(7)Outcomes:

(i) Mortality: Overall, 10 of the 16 included studies reported the outcome of mortality [25,27,33,35,36,37,39,43,44,45]. Mortality was reduced in the intervention group but it was not statistically significant in the individual studies [25,27,33,36,37,39,43,45]. On meta-analysis there was a significant reduction in mortality with an event rate of 9% in the plant-derived product group versus 20% in the control group [12/134 vs. 27/135; RR: 0.48 (95% CI: 0.26–0.87); *p* = 0.02; I^2^ = 0%] (Figure 2).

(ii) Histological NEC: All the 16 included studies reported improvement in the microscopic findings related to NEC. A significant reduction specifically in histological scores after intervention with plant-derived substances were reported by 11 studies [25,27,28,33,34,35,39,40,43,44,45]. Six studies reported the data in a format suitable for pooling [35,36,37,41,42,44]. Five out of these six studies reported a significant improvement in number of animal with no NEC or NEC with reduced severity in the intervention group [25,36,37,41,42]. On meta-analysis, there was a significant reduction in histological NEC with an event rate of 19% in the plant-derived product group versus 70% in the control group: [24/126 vs. 55/79; RR: 0.34 (95% CI: 0.22–0.52); *p* < 0.001; I^2^ = 8.8%] (Figure 3).

(8)Other outcomes:

(i) Clinical assessment of the severity of NEC: Six [25,27,33,39,43,45] out of nine studies that reported the clinical severity of NEC [25,26,27,33,37,39,43,44,45] showed statistically significant improvement in the intervention group. The remaining three studies also reported clinical improvement, but it was not statistically significant [37,40,44].

(ii) Laboratory markers of inflammation and oxidative injury: Markers of inflammation were reported in 11 studies [25,28,33,34,36,38,40,41,42,43,45] and markers of oxidative injury were reported in 13 studies [25,27,28,33,34,35,36,37,39,41,43,44,45]. All studies showed an improvement in markers of inflammation and oxidative stress. There were four RCTs that reported only descriptive data and data on the mechanism of action of the plant-derived substance studied [28,34,38,40]. Evidence on apoptosis or pyroptosis was reported in nine studies [27,28,33,35,38,39,43,44,45] and evidence on integrity of the intestinal barrier was reported in one study [45]. All studies reported decreased apoptosis and improved intestinal integrity.

Publication bias: Publication bias was assessed only for the outcomes of mortality since it had ≥10 studies in the meta-analysis. Visual inspection of the funnel plot (Figure 4) and the results of Begg’s test, Egger’s test, and Harbord’s test suggested that publication bias was unlikely for mortality (Egger’s test *p* value: 0.0.86, Begg’s test *p* value: 0.36, and Harbord’s test *p* value: 0.36).

Certainty of evidence: The level of evidence was deemed low when considering small sample size, high ROB of the included studies, and significant heterogeneity.

## 4. Discussion

Our systematic review showed that supplementation with plant-derived products significantly reduced mortality and severity of histologically diagnosed NEC in rodent models of illness. Findings for NEC were consistent across studies despite the heterogeneity of products, dose, route of administration, and duration of supplementation.

Overall, our findings suggest that as a class of intervention, plant-derived substances may have shared pathways of benefit (e.g., anti-inflammatory and antioxidant properties) for reducing the incidence and severity of NEC. This hypothesis is supported by the histopathological findings of reduced severity of gut injury, along with decreased expression of various pro-inflammatory mediators, reduced total oxidant score, oxidative stress index, and increased total antioxidant score. The histopathological findings correlate well with the current understanding of the pathogenesis of NEC that involves an exaggerated pro-inflammatory response, decreased anti-inflammatory response, reactive oxygen species, and oxidative stress [1,2,7,8,9,10,11,47]. Targeting these pathways, as plant-derived substances assessed in the included studies appear to do so, could be fundamental in modifying the disease process.

The benefits of plant-derived substances relate to their phytochemical content which increase the quantity and diversity of gut microbiota, reduce inflammation and oxidative stress, enhance immune function, regulate gene expression and intracellular signalling, and reduce DNA damage [48,49]. Evidence supports their role in the prevention of NEC through trophic effects on intestinal mucosa and increased secretion of regulatory peptides and digestive enzymes resulting in accelerated intestinal maturation [50]. The discovery of the anti-inflammatory and anti-oxidant properties of plant-derived substances is not a new finding. Based on these properties, such products have been used for centuries, to promote health and treat various conditions including communicable diseases, cardiovascular disease, inflammatory conditions, and cancers [15,16,51,52]. Our findings open a new frontier for developing novel therapeutic agents using plant-derived substances for the prevention and treatment of NEC.

A comprehensive review of all ingredients included in our review is beyond the scope of this report. Evidence suggests that curcumin, ginger and fennel (all considered generally safe by the FDA, USA (https://www.accessdata.fda.gov/scripts/cdrh/cfdocs/cfcfr/CFRSearch.cfm?fr=582.10&SearchTerm=fennel, accessed on 12 June 2023)) warrant further evaluation in high quality pre-clinical studies. Safe use of ginger and curcumin in pregnant or lactating women and in the paediatric age group is reassuring [53,54,55,56,57,58]. Comprehensive search for the evidence on the safety and efficacy, and independent assessment of the quality of a plant-derived substance/product is critical if it is to be developed for use in preterm infants.

The limitations of our review need to be acknowledged. These include the small sample size of the included individual studies, high ROB due to the unclear method for allocation concealment, sub-optimal blinding in majority of the included studies, lack of details of registration of the study protocol on an animal trial registry, and heterogeneity in the methods used for inducing NEC and evaluated plant-derived substances. The possibility exists of shared mechanisms of benefits (e.g., anti-inflammatory effect, and protection from free radicle injury) of plant-derived substances as a class of intervention. We hence conducted a meta-analysis for the outcomes of NEC and mortality. However, future research should focus separately on each plant-derived substance as their effects could be specific to the plant/ingredient.

To our knowledge, this is the first comprehensive systematic review and meta-analysis in this field with robust methodology using SYRCLE’s ROB tool for animal studies. Despite the limitations, we believe that our findings add meaningful data in an important field of research in neonatal medicine, considering the reasonably large total sample size, use of the validated preterm rat pup model of NEC, and consistent significant benefits for mortality and NEC across studies with narrow confidence intervals and low *p* values.

Our findings are beneficial in designing and conducting robust, definitive, and adequately powered RCTs in validated models of NEC including preterm rat pups to ascertain the safety and efficacy of the plant-derived substances assessed in the studies included in our review. The evidence provided in our review is helpful in selecting the most appropriate plant-derived substances for further evaluation in pre-clinical studies. A note of caution is warranted in selection of products for further translational research considering that the safety and efficacy of plant-derived substances cannot be ensured given the poor regulatory control over the herbal medicine and food supplement industry [59,60]. Certain plants used for medicinal purposes have been shown to be harmful [59,60].

It is important to note that results from animal models may not translate into clinical benefits as the effects of an intervention could be species-specific with significant differences in pharmacokinetics of therapeutic agents between humans and animal species [61,62,63,64]. Furthermore, different exposures (e.g., hypoxia, hypothermia, LPS, formula feeding, acetic acid enema) inducing the illness in laboratory animals may not result in NEC as it occurs in its complex and multifactorial nature in human preterm infants [61,62,63].

## 5. Conclusions

In summary, the results of our systematic review indicate the potential of plant-derived substances as a novel option in the prevention and treatment of NEC in preterm infants. Rigorous evaluation of the most suitable agent/s selected from the studies included in our review is needed for further progress in this field.

## Figures and Tables

**Figure 1 nutrients-16-00832-f001:**
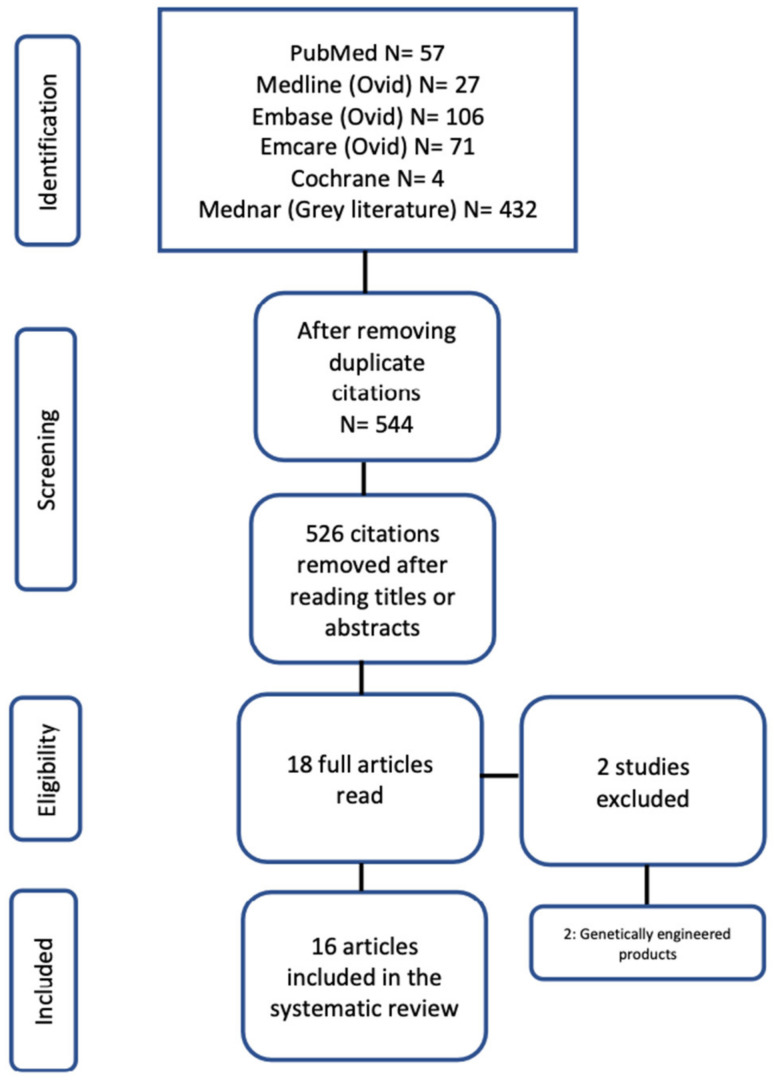
PRISMA flow chart for study selection.

**Figure 2 nutrients-16-00832-f002:**
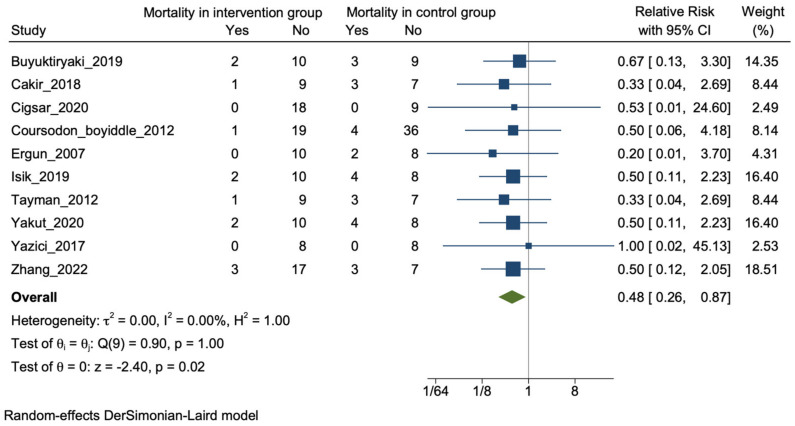
Forrest plot for the outcome of mortality in the included studies [25,27,33,35,36,37,39,43,44,45].

**Figure 3 nutrients-16-00832-f003:**
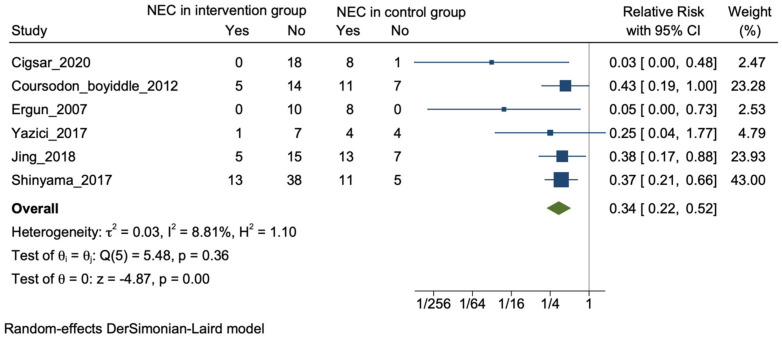
Forrest plot for the outcome of NEC histology in the included studies [35,36,37,41,42,44].

**Figure 4 nutrients-16-00832-f004:**
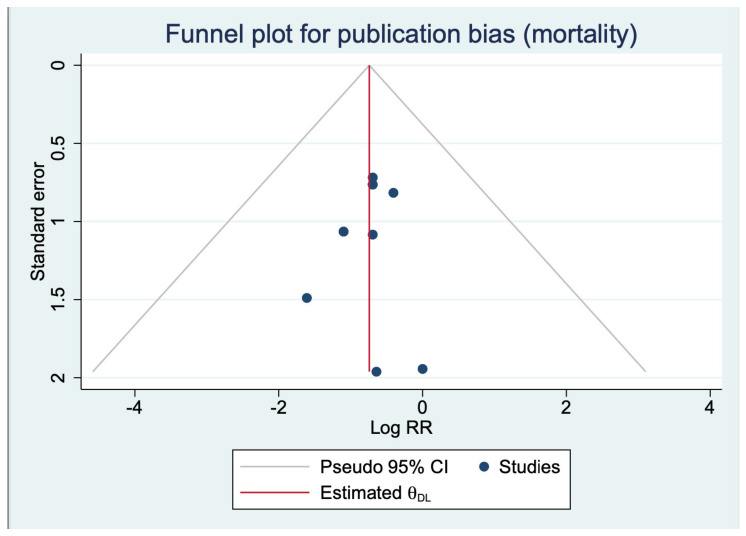
Funnel plot for the outcome of mortality.

**Table 1 nutrients-16-00832-t001:** Characteristics of the included studies.

Study ID	Intervention/Predominant Phytochemicals/Methods	Outcomes						Author’s Conclusions
		Mortality	Clinical Illness Score	Macroscopic Assessment Score	Histological Injury	Inflammatory and Oxidative Stress	Other	
Zhang (2022) [45] China	Intervention: Oral *Rhus chinensis* Predominant Phytochemicals: Gallic acid (48.78%) Quercetin-3-O-rhamnoside (27.54%). Sample: Neonatal Sprague Dawley rat pups (*n* = 40) Gr 1 (*n* = 10): Breastfeeding Cs; Gr 2 (*n* = 10): NEC protocol; Gr 3 (*n* = 10): NEC protocol + low dose *Rhus chinensis* extract (200 mg/kg); Group 4 (*n* = 10): NEC protocol + high dose *Rhus chinensis* extract (400 mg/kg). NEC Induction: Formula fed, hypoxia-reoxygenation and hypothermia. Rats euthanised: Day 4	Group 1: 0/10 Group 2: 3/10 (30%) Group 3: 2/10 (20%) Group 4: 1/10 (10%). ^a^	Gr 1: 0; Gr 2: 7.71 +/− 0.57; Gr 3: 5.46 +/− 0.45; Gr 4: 4.50 +/− 0.54. The decline in clinical sickness score between Gr 2 and Gr 3 was 29.18% and between Gr 2 and Gr 4 was 41.63% (*p* < 0.05). ^a^	Gr 1: 0; Gr 2: 4.93 +/− 0.20; Gr 3: Exact figures not reported; Group 4: 2.86 (SD not reported). Macroscopic assessment score declined by 41.99% between Gr 2 and Gr 4 (*p* < 0.05). ^a^	Gr 1: 0; Gr 2: 3.17 +/− 0.31; Gr 3: 2.14 +/− 0.26; Gr 4: 1.50 +/− 0.19. The intestinal injury score was 32.49% and 52.68% lower in Gr 3 and Gr 4, respectively compared to Gr 2 (*p* < 0.05). ^a^	Inflammatory markers: ↑ TNF-α, IL-6, TLR4, NF-κβ and p-NF but ↓ NRF2 in Gr 2 compared with Gr 1, Gr 3 and Gr 4 (*p* < 0.05). Markers of oxidative stress: ↑ TOS and MDA and ↓ TAS, SOD and GSH-Px in Gr 2 compared with Gr 1 (*p* < 0.05). ↑ TOS and MDA and ↓ TAS in Gr 2 compared to Grs 3 and 4; ↓ SOD and GSH levels in Gr 2 compared to Gr 4. ↑ MPO, iNOS and ↓ NQO1 in Gr 2 compared with Gr 1, Gr 3 and Gr 4 (*p* < 0.05).	Intestinal integrity: Levels of ZO-1 and Occludin higher in Grs 1, 3 and 4 compared with Gr 2 (*p* < 0.05). Markers of apoptosis: ↑ TUNEL-positive signals in Gr 2 compared with Grs 1, 3 and 4 (*p* < 0.05). ↑ expression of cleaved Caspase-3 and Bax and ↓ levels of Bcl-2 in Gr 2 compared with Grs 1, 3 and 4 (*p* < 0.05).	*Rhus chinensis* extract, particularly at high doses, has a protective effect against NEC in a rat NEC model. This is likely due to maintenance of intestinal barrier integrity and inhibition of oxidative stress, inflammation, and apoptosis.
Yin (2020) [28] China	Intervention: Curcumin orally Predominant Phytochemicals: NA Sample: Newborn rat pups (*n* = 20) Gr 1 (*n* = 5): Cs; Gr 2 (*n* = 5): NEC protocol; Gr 3 (*n* = 5): NEC protocol + low dose curcumin 20 mg/kg (NEC + LDI); Gr 4 (*n* = 5): NEC protocol + high dose curcumin 50 mg/kg (NEC + HDI). NEC Induction: Hypoxia-reoxygenation and hypothermia. Rats euthanised: Not reported	NA	NA	NA	Exact scores not given. Scores were highest in the NEC gr followed in order by NEC + LDI gr, NEC + HDI gr and no injury in the C gr (*p* < 0.01).	Inflammatory markers: IL-1β, IL-6, IL-18 and TNF-α were highest in NEC gr followed in order by NEC + LDI, NEC + HDI and C grs (*p* < 0.01). Regulation of inflammation: ↓ levels of SIRT1 and NRF2 in NEC gr compared with C, NEC + LDI and NEC + HDI grs (*p* < 0.01). ↑ expression of TLR4 in NEC gr compared with C, NEC + LDI and NEC + HDI (*p* < 0.01). Curcumin inhibited release of inflammatory factors IL-1β, IL-6, IL-18 and TNF-α after activation with LPS/ATP.	Inhibition of pyroptosis: Curcumin treatment upregulated expression of SIRT1 and NRF2 and down-regulated expression of TLR4, NLRP3 and cleaved caspase-1 compared to NEC gr (*p* < 0.01). Protective function of curcumin was lost after inhibition with sirtinol.	Curcumin improves the inflammatory state in NEC, inhibits the expression of inflammatory mediators, reduces inhibition of SIRT1 /NRF2 pathway in NEC, and inhibits TLR4 expression.
Yakut (2020) [43] Turkey	Intervention: Fennel extract Predominant Phytochemicals: Curmin, kaempferol, vanillic acid, hydroxybenzoic acid and salicylic acid. Sample: Neonatal Wistar rat pups (*n* = 36). Gr 1: Breastfed controls (*n* = 12), Gr 2: NEC (*n* = 12); Gr 3: NEC + fennel extract (*n* = 12). NEC Induction: 1 mg/kg LPS orally, hyperosmolar formula, hypoxia-reoxygenation and cold stress. Rats euthanised: Day 4.	Gr 1: 0/12; Gr 2: 4/12; Gr 3: 2/12	Gr 1: 0; Gr 2: 8 (2); Gr 3: 5 (2) (*p* < 0.001). ^b^	Gr 1: 0; Gr 2: 6 (2); Gr 3: 3 (1) *(p* = 0.013) ^b^	Gr 1: 0; Gr 2: 4 (2); Gr 3: 2 (1) (*p* < 0.001) ^b^	Inflammatory markers: IL-6 and TNF-α levels were ↑ in gr 2 vs. gr 3 (*p* < 0.001). Markers of oxidative stress: TOS, OSI, AOPP, LPO, 8-OHdG and MPO levels were ↑ in gr 2 vs. gr 3 (*p* < 0.001). TAS levels were significantly ↑ in gr 3 vs. gr 2 (*p* < 0.001).	Evidence of apoptosis: Lower levels of caspase-3, -8 and -9 in gr 3 vs. gr 2 (*p* < 0.001).	Fennel prevents and reduces damage from NEC in an experimental rat model through its anti-oxidant, cytoprotective, and anti-inflammatory properties.
Cigsar (2020) [35] Turkey	Intervention: IP and oral sesamol Predominant Phytochemicals: NA Sample: Neonatal Wistar rats (*n* = 34). Gr 1 (*n* = 9): NEC model; Gr 2 (*n* = 9): NEC + IP sesamol; Gr 3 (*n* = 9): NEC + oral sesamol; Gr 4 (*n* = 7): Healthy controls. NEC induction: Hypoxia-hyperoxia-cold stress and hyperosmolar formula feeds. Rats euthanised: Day 4.	No pups died	NA	NA	Gr 1: 8/9, mean 2.78 +/− 0.97; Gr 2: 0/9, mean 0.33 +/− 0.5; Gr 3: 0/9, mean 0.94 +/− 1.2; Gr 4: 0/7, mean 0.14 +/− 0.	Markers of oxidative stress: MDA levels were ↑ in gr 1 vs. grs 2, 3 and 4 (*p* < 0.01). SOD and GSH-Px were ↓ in gr 1 vs. gr 4 but the differences between gr 1 and grs 2 and 3 did not meet clinical significance.	Evidence of apoptosis: Fewer Bcl-2 and caspase-3 positive cells in grs 2 and 3 vs. gr 1 (*p* < 0.001).	Sesamol appears to reduce the intestinal damage in NEC through its anti-inflammatory, antioxidant, antiapoptotic, and cytoprotective effects.
Buyuktiryaki (2019) [33] Turkey	Intervention: IP Ankaferd blood stopper Predominant Phytochemicals: NA Sample: Neonatal Wistar rats (*n* = 36). Gr 1: Healthy breastfed controls (*n* = 12); Gr 2: NEC (*n* = 12); Gr 3: NEC + IP ABS (*n* = 12). NEC induction: IP injection of 1 mg/kg LPS, formula feeding and hypoxia-hyperoxia-cold stress. Rats euthanised: Day 4.	Gr 1: 0/12; Gr 2: 3/12; Gr 3: 2/12 (*p* = 0.53). ^b^	Gr 1: 0; Gr 2: 7 +/− 1; Gr 3: 4 +/− 1 (*p* <0.001). ^b^	Gr 1: 0; Gr 2: 5.5 +/− 2.25; Gr 3: 3.0 +/− 1 (*p* = 0.01). ^b^	Gr 1: 0; Gr 2: 3.5 +/−1 Gr 3: 2 +/−1 (*p* = 0.001)	Inflammatory markers: TNF-α and IL-1β were ↓ in the NEC + ABS gr vs. NEC gr (*p* < 0.001). Markers of oxidative stress: ↓ TOS, LPO, AOPP, 8-OHdG and OSI in NEC + ABS vs. NEC group (*p* < 0.001). GSH, SOD and TAS levels were ↑ in NEC + ABS vs. NEC gr (*p* < 0.05).	Evidence of apoptosis: Fewer cells positive for caspase-3, -8, and -9 in the NEC + ABS gr vs. NEC gr (*p* = 0.001).	ABS protects against intestinal damage in an experimental rat model of NEC by exerting anti-inflammatory, antiapoptotic, and antioxidant effects.
Isik (2019) [39] Turkey	Intervention: Oral sumac (Rhus coriaria) Predominant Phytochemicals: Vanillic acid, catechinhyrate, myricetin, gallic acid, ellagic acid Sample: Neonatal rats (*n* = 36). Gr 1: Breastfed Cs (*n* = 12); Gr 2: NEC protocol (*n* = 12); Gr3: NEC + Rhus coriaria (*n* = 12). NEC induction: Formula feeding, hypoxia-reoxygenation-cold stress. Rats euthanised: Day 4.	Gr 1: 0; Gr 2: 4/12 Gr 3: 2/12 (*p* = 0.09)	Gr 1: 0; Gr 2: 7.0 (0.5) Gr 3: 4.0 (1.0) (*p* < 0.001) ^b^	Group 1: 0; Group 2: 5 (1); Group 3: 2 (1.5); (*p* = 0.015) ^b^	Group 1: 0; Group 2: 3.5 (1.0); Group 3: 2.0 (1.0); (*p* = 0.002) ^b^	Markers of oxidative stress: Tissue OSI, 8-OHdG and LPO were ↓ in intervention vs. NEC grs. TAS higher and TOS and AOPP lower in intervention vs. NEC gr.	Evidence of apoptosis: Fewer cells positive for caspase-3, -8, and -9 in the NEC + Rhus coriaria gr vs. NEC gr (*p* < 0.05).	Sumac prevents intestinal injury in a rat NEC model due to its anti-inflammatory, antioxidant, immunomodulatory, and anti-apoptotic properties.
Fang (2018) [38] China	Intervention: Oral berberine Predominant Phytochemicals: NA Sample: Mouse pups (*n* = 30). Gr 1: NEC protocol (*n* = 15); Gr 2: NEC + berberine (*n* = 15) NEC induction: Acetic acid enema (150 mmol/L). Rats euthanised: Day 10	NA	NA	NA	Berberine decreased area of infarction in berberine vs. control gr (no measure of statistical significance provided).	Inflammatory markers: TLR4, MD-2, TNF-α, NF-κβ, IL-6 and Cxcl-1 in peripheral blood were ↓ in intervention gr compared to NEC gr.	Evidence of apoptosis: Caspase-3 and -9 were ↓ in NEC + berberine gr vs. NEC gr. P13K/AKT signalling pathway: Berberine treatment inhibited expression and phosphorylation of P13K and AKT in epithelial cells vs. Cs. (*p* < 0.01).	Berberine reduces epithelial cell apoptosis and tissue necrosis by inhibiting inflammation and downregulating PI3K/AKT
Jing (2018) [41] China	Intervention: Oral berberine Predominant Phytochemicals: NA Sample: Newborn Sprague Dawley rats (*n* = 60). Gr 1: Breastfed Cs (*n* = 20); Gr 2: NEC protocol (*n* = 20); Gr 3: NEC + berberine (*n* = 20). NEC induction: Intermittent hypoxia and oral LPS. Rats euthanised: At 96 h.	NA	NA	NA	Gr 1: 0; Gr 2: 13/20; Gr 3: 5/20; (*p* < 0.05)	Inflammatory markers: TLR4 mRNA, NF-kB, IL-6, IL-10 and TNF-α mRNA significantly ↓ in intervention gr compared to NEC gr. MUC2 and SIgA ↑ in intervention compared to NEC gr. Markers of oxidative stress: iNOS significantly ↓ in intervention gr compared to NEC gr.	NA	Enteral administration of berberine ameliorates the clinical symptoms and decreases the incidence of NEC in a neonatal rat model by down-regulating TLR4 thereby inhibiting production of inflammatory mediators and upregulating expression of MUC2 and SIgA.
Cakir (2018) [25] Turkey	Intervention: Ginger Predominant Phytochemicals: Vanillic acid, fumaric acid, resveratrol, curmin, silymarin. Sample: Newborn Wistar rats (*n* = 30). Gr 1: NEC protocol (*n* = 10); Gr 2: NEC + ginger (*n* = 10); Gr 3: Breastfed Cs (*n* = 10). NEC induction: Intraperitoneal LPS, formula feeding, hypoxia-reoxygenation-cold stress. Rats euthanised: Day 4.	Gr 1: 3/10; Gr 2:1/10; Gr 3: 0	Gr 1: 8.1 (2.1); Gr 2: 4.8 (0.5); Gr 3: 0; (*p* = 0.013). ^b^	Gr 1: 5.0 (2.0); Gr 2: 3.0 (0.5); Gr 3: 0; (*p* = 0.032). ^b^	Gr 1: 2.7 (0.2); Gr 2: 1.5 (0.3); Gr 3: 0; (*p* = 0.017). ^b^	Inflammatory markers: TNF-α, IL-1β and IL-6 in lung tissue ↓ in NEC + ginger gr vs. NEC gr (*p* < 0.05). Markers of oxidative stress: Tissue GSH-Px and SOD ↑ NEC + ginger gr vs. NEC gr (*p* < 0.05). Tissue MDA, MPO and XO ↓ in NEC + ginger gr vs. NEC gr.	NA	Ginger protects intestinal tissues from severe damage in NEC due to its antioxidant, anti-inflammatory, anti-apoptotic and immunomodulatory properties.
Yazici (2017) [44] Turkey	Intervention: Quercetin Predominant Phytochemicals: NA Sample: Sprague Dawley rat pups (*n* = 24). Gr 1: Breastfed controls (*n* = 8); Gr 2: NEC protocol (*n* = 8); Gr 3: NEC + quercetin (*n* = 8). NEC induction: Hypoxia-reoxygenation-cold stress. Rats euthanised: At 72 h.	No deaths	Slight hypotonia and hypoactivity noted in NEC gr vs. other grs on day 3.	NA	Gr 1: 0/8; Gr 2: 4/8, median 1.5 Group 3: 1/8, median 0.5 (*p* < 0.05) ^b^	Markers of oxidative stress: Increased TAS in intervention gr vs. NEC + healthy Cs. ↓ TOS in intervention vs. NEC gr, ↓ MDA levels in intervention vs. NEC but similar to healthy controls. ↑ GSH in intervention vs. NEC but similar to healthy Cs.	Evidence of apoptosis: Apoptotic index: Gr 1: 2.7 (1.2); Gr 2: 7.0 (1.9); Gr 3: (2.5 (0.4); (*p* < 0.05).	Strong correlation between Quercetin treatment and reduction in oxidative stress. Quercetin pre-treatment appears to temper the decrease in GSH, inhibits increases in lipid peroxidation and TOS, and increases TAS. Protective effect on NEC development.
Shinyama (2017) [42] Japan	Intervention: Daikenchuto Predominant Phytochemicals: NA Sample: Sprague Dawley rat pups (*n* = 67). Group 1: NEC controls (*n* = 16); Group 2: NEC + 0.3 g/kg/day daikenchiuto (*n* = 10); Group 3: 0.6 g/kg/day daikenchuto (*n* = 26); Group 4: NEC + 1.0 g/kg/day daikenchuto (*n* = 15). NEC induction: Formula fed, hypoxia-reoxygenation + LPS Rats euthanised: Day 5.	NA	NA	NA	Gr 1: NEC in 11/16; Gr 2: NEC in 3/10; Gr 3: 8/26 Gr 4: 2/15 (*p* = 0.019 for gr 1 vs. gr 4).	Inflammatory markers: High-dose Daikenchuto ↓ IL-6 in gr4 vs. gr 1 (*p* = 0.04).	NA	Administration of high-dose Daikenchuto improved incidence of NEC in a rat model of NEC, possibly via the suppression of IL-6.
Cai (2016) [34] China	Intervention: Astragaloside IV Predominant Phytochemicals: NA Sample: Sprague Dawley rat pups (*n* = 40). Gr 1 (*n* = 10): NEC control rats given saline; Gr 2 (*n* = 10): NEC rats given 25 mg/kg/d AS-IV; Group 3 (*n* = 10): NEC rat given 50 mg/g/d AS-IV; Gr 4 (*n* = 10): NEC rats treated with 75 mg/kg/d AS-IV. NEC induction: Hypoxia-reoxygenation, hypothermia. Rats euthanised: Day 4.	NA	NA	NA	Exact scores not reported. Gr 1: 2.6 +/− 0.53; Gr 2: 2.1 +/− 0.83; Gr 3: 1.9 +/− 0.87; Gr 4: 1.4 +/− 0.51 (*p* < 0.05 for both gr 3 and gr 4 compared with gr 2).	Inflammatory markers: Treatment with AS-IV at 50 mg/kg/d and 75 mg/kg/d resulted in significantly ↓ serum TNF-α (*p* = 0.042 and *p* = 0.0112, respectively), ↓ IL-1β (*p* = 0.0086 and *p* = 0.0065, respectively) and ↓ IL-6 (*p* = 0.0243 and *p* = 0.0083, respectively). mRNA expression of TNF-α, IL-1β, IL-6 and NF-κB significantly ↓ in AS-IV treated rats Markers of oxidative stress: GSH levels were significantly ↑ in the AS-IV treated groups than NEC group (*p* = 0.0094). SOD and MDA levels were significantly ↓ in AS-IV treated rats.	NA	Treatment with AS-IV may protect the intestine against NEC by modulating the oxidative/antioxidative balance, reducing inflammatory mediators.
Tayman (2012) [27] Turkey	Intervention: Nigella sativa oil Predominant Phytochemicals: NA Sample: Sprague Dawley rat pups (*n* = 30). Gr 1: NEC protocol (*n* = 10); Gr 2: NEC + NSO (*n* = 10); Gr 3: Breastfed controls (*n* = 10). NEC induction: Formula fed, hypoxia-reoxygenation-cold stress. Rats euthanised: Day 4.	Gr 1: 3/10 Gr 2: 1/10 Gr 3: 0; (*p* = 0.28)	Gr 1: 7.4 (2.2); Gr 2: 4.2 (0.8); Gr 3: 0; (*p* = 0.024) ^a^	Group 1: 5.0 (3.0); Group 2: 2.0 (0.5); Group 3: 0 (*p* = 0.036) ^a^	Group 1: 2.6 (0.3); Group 2: 1.6 (0.2); Group 3: 0 (*p* = 0.015) ^a^	Markers of oxidative stress: GSH-Px and SOD levels preserved in intervention group (*p* < 0.05). Lower levels of MDA, MPO + XO in intervention group. (*p* < 0.05).	Evidence of apoptosis: Apoptosis score: Gr 1: 2.8 (0.6); Gr 2: 1.8 (0.5); Gr 3: 0 (*p =* 0.003)	NSO may be a promising agent for protecting intestinal tissue from NEC due to its antioxidant, anti-inflammatory, anti-apoptotic and immunomodulatory effects.
Coursodon-Boyiddle [36] (2012) USA	Intervention: Oral pomegranate seed oil Predominant Phytochemicals: NA Sample: Neonatal Sprague Dawley rats (*n* = 60). Gr 1: Breastfed Cs (*n* = 20); Gr 2: NEC protocol (*n* = 20); Gr 3: NEC + PSO (*n* = 20). NEC induction: Hypoxia-reoxygenation-cold stress. Rats euthanised: At 96 h.	Group 1: 2/20; Group 2: 2/20; Gr 3: 1/20	NA	NA	Gr 1: 0/18; Gr 2: 11/18; Gr 3: 5/19; (*p* < 0.01 gr 2 vs. gr 3)	Inflammatory markers: ↓ IL-6, IL-8 and TNF-α expression in PSO vs. NEC group (*p* < 0.05).		Orally administered PSO protects against NEC in the rat model as evidenced by increased enterocyte proliferation, protection of intestinal cell architecture + ↓ inflammatory response at the site of injury (the ileum).
Jia (2010) [40] China	Intervention: IP curcumin Predominant Phytochemicals: NA Sample: Rat pups (*n* = 40). Gr 1: Breastfed Cs + IP injection of saline (*n* = 10); Gr 2: Breastfed controls + IP injection of solvent; Gr 3: NEC protocol (*n* = 10); Gr 4: NEC + IP injection of curcumin. NEC induction: Hypoxia-reoxygenation and cold stress. Rats euthanised: Day 4.	NA	Described as increased severity of illness in NEC vs. intervention group. Not scored.	Described as increased severity in NEC vs. intervention group. Not scored.	Gr 1: 0.33 +/−0.35; Gr 2: 0.37 +/− 0.43; Gr 3: 3.2 +/− 0.57; Gr 4: 1.6 +/− 0.44 (*p* < 0.01) ^a^	Inflammatory markers: TNF-α lower and IL-10 ↑ in intervention group vs. NEC group (*p* < 0.05). Markers of oxidative stress: COX-2 expression in intestinal tissue significantly ↓ in intervention group (*p* < 0.05)		Curcumin improves general condition, reduces intestinal damage, reduces pro-inflammatory mediators, and increases anti-inflammatory mediators in a rat NEC model.
Ergun (2007) [37] Turkey	Intervention: Oral resveratrol Predominant Phytochemicals: NA Sample: Wistar rat pups (*n* = 27). Gr 1: Breastfed Cs (*n* = 7); Gr 2: NEC protocol (*n* = 10); Gr 3: NEC + resveratrol (*n* = 10). NEC induction: Hypoxia-reoxygenation and formula fed. Rats euthanised: Day 4.	Gr 1: 0/7; Gr 2: 2/10; Gr 3: 0/10.	Described as ↑ severity of illness in NEC vs. intervention gr. Not scored.	Described as increased severity in NEC vs. intervention gr. Not scored.	Gr 1: 0/7; Gr 2: 8/10; Gr 3: 0/10.	Markers of oxidative stress: Ileal tissue nitrate/nitrite levels for groups 1, 2 and 3 were 178.3 ± 7, 191.4 ± 4.1, and 181 ± 3.6 μmol/(L·g), respectively (*p* < 0.01). Intestinal iNOS expression was ↑ in the intervention (*p* < 0.01).		Enteral resveratrol attenuates iNOS expression and therefore NO-mediated nitrosative stress, and preserves intestinal epithelial integrity and mucosal barrier.

NEC: Necrotising enterocolitis; a: Mean and standard deviation; b: Median and interquartile range; TOS: Total oxidant score; MDA: Malondialdehyde; TAS: Total antioxidant score; SOD: Superoxide dismutase; GSH-Px: Glutathione peroxidase; GSH: Glutathione; MPO: Myeloperoxidase; TNF-α: Tumor necrosis factor-α; IL-6: Interleukin-6; NRF2: Nuclear factor erythroid 2-related factor; NQO1: NADH (Nicotinamide Adenine Dinucleotide plus Hydrogen) Quinone Oxidoreductase 1; TLR4: Toll-like receptor-4; NF-κβ: Nuclear factor-κβ; p-NF: Precursor to nuclear factor; iNOS: Inducible nitric oxide synthase; ZO-1: Zonula occludens-1; TUNEL: Terminal deoxynucleotidyl transferase dUTP nick end labelling; Bax: Bcl-2 associated x; Bcl-2: B-cell lymphoma 2; IL-1β: Interleukin-1β; IL-18: Interleukin-18; SIRT1: Silent information regulator 2-related protein 1; LPS: Lipopolysaccharide; ATP: Adenosine triphosphate; NLRP3: NOD-, LRR- and pyrin domain-containing protein 3; OSI: Oxidative stress index; AOPP: Advanced oxidation protein products; LPO: Lipid hydroperoxide; 8-OHdG: 8-Hydroxydeoxyguanosine; MD-2: Myeloid differentiation protein-2; Cxcl-1: Chemokine ligand-1; P13K: Phosphoinositide 3-kinase; AKT: Protein kinase B; MUC2: Mucin-2; SIgA: RNA polymerase σ factor SigA; XO: Xanthine oxidase; JAM-A: Junctional Adhesion Molecule A. IP: Intraperitoneal; NA: Not available; Gr: Group, SD: Standard deviation; C: Control; ABS: Ankaferd blood stopper; HDI: High dose intervention; LDI: Low dose intervention; ↑: Increased; ↓: Decreased.

**Table 2 nutrients-16-00832-t002:** Risk of Bias assessment for included studies using SYRCLE (SYstematic Review Centre for Laboratory animal Experimentation) tool.

Study ID	Sequence Generation	Baseline Characteristics	Allocation Concealment	Random Housing	Blinding Caregivers and Researchers	Blinding of Outcome Assessors	Random Outcome Assessment	Incomplete Outcome Data ^α^	Selective Reporting ^β^	Other Sources of Bias (e.g., Estimation of Sample Size)
Zhang [45] (2022) China	Unclear	Unclear	Unclear	Unclear	Unclear	Low risk for clinical sickness scores (blinded). Unclear for other outcomes (blinding not discussed)	Unclear	Unclear	Unclear	Unclear
Yin (2020) [28] China	Unclear	Unclear	Unclear	Unclear	Unclear	Unclear	Unclear	Unclear	Unclear	Unclear
Yakut (2020) [43] Turkey	Unclear	High risk: Baseline details provided but not per appropriately randomized groups.	Unclear	Unclear	Unclear	Low risk for clinical sickness score (blinded) Unclear for other outcomes (blinding not discussed)	Unclear	Unclear	Unclear	Unclear
Cigsar (2020) [35] Turkey	Unclear	Unclear	Unclear	Unclear	Unclear	Low risk (blinded to clinical and histopathological outcomes)	Unclear	Unclear	Unclear	Unclear
Buyuktiryaki (2019) [33] Turkey	Unclear	High risk: Baseline details provided but not per appropriately randomized groups.	Unclear	Unclear	Unclear	Low risk (blinded to clinical and histopathological outcomes)	Unclear	Unclear	Unclear	Unclear
Isik (2019) [39] Turkey	Unclear	High risk: Baseline details provided but not per appropriately randomized groups.	Unclear	Unclear	Unclear	Low risk (blinded to clinical and histopathological outcomes)	Unclear	Unclear	Unclear	Unclear
Fang (2018) [38] China	Unclear	Unclear	Unclear	Unclear	Unclear	Unclear	Unclear	Unclear	Unclear	Unclear
Jing (2018) [41] China	High risk (randomized but unclear how)	Unclear	Unclear	Unclear	Unclear	Unclear	Unclear	Unclear	Unclear	Unclear
Cakir (2018) [25] Turkey	Unclear	High risk: Baseline details provided but not per appropriately randomized groups.	Unclear	Unclear	Unclear	Low risk for clinical sickness scores (blinded) Unclear for other outcomes (blinding not discussed	Unclear	Unclear	Unclear	Unclear
Yazici (2017) [44] Turkey	Unclear	High risk: Baseline details provided but not per appropriately randomized groups.	Unclear	Unclear	Unclear	Low risk (blinded for histopathological outcomes; clinical outcomes not assessed).	Unclear	Unclear	Unclear	Unclear
Shinyama (2017) [42] Japan	Unclear	Unclear	Unclear	Unclear	Unclear	Unclear	Unclear	Unclear	Unclear	Unclear
Cai (2016) [34] China	Unclear	Unclear	Unclear	Unclear	Unclear	Unclear	Unclear	High risk: Outcome data presented for 7/10 rats in group 1 and 8/10 rats in group 2. No explanation given for missing data.	Unclear	Unclear
Tayman (2012) [27] Turkey	Unclear	High risk: Baseline details provided but not per appropriately randomized groups.	Unclear	Unclear	Unclear	Low risk (blinded for histopathological outcomes; clinical outcomes not assessed)	Unclear	Unclear	Unclear	Unclear
Coursodon-Boyiddle (2012) [36] USA	Unclear	Unclear	Unclear	Unclear	Unclear	Low risk (blinded to clinical and histopathological outcomes)	Unclear	Unclear	Unclear	Unclear
Jia (2010) [40] China	Low risk	Low risk	Unclear	Unclear	Unclear	Unclear	Unclear	Unclear	Unclear	Unclear
Ergun (2007) [37] Turkey	Unclear	High risk: Baseline details provided but not per appropriately randomized groups.	Unclear	Unclear	Unclear	Low risk (blinded to clinical and histopathological outcomes)	Unclear	Low risk	Unclear	Unclear

Incomplete outcome data ^α^: Risk of bias considered unclear if study protocol not registered in a registry for animal studies as it was not possible to check outcomes reported with aims of original study protocol; Selective reporting ^β^: Risk of bias considered unclear if study protocol not entered in a registry for animal studies as it was not possible to check outcomes reported with aim and methodology of original study protocol.

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
