# Peer review of "Plant-Derived Substances for Prevention of Necrotising Enterocolitis: A Systematic Review of Animal Studies"

_nutrients, 2024, doi:10.3390/nu16060832_

Round 1

Reviewer 1 Report

Comments and Suggestions for Authors

The systematic review by Mackay and colleagues reports the effect of plant-derived substances in preventing necrotising enterocolitis. The review paper is well-architectured and the methodological approach based on prisma protocol is well thought out.

As reported in the discussion section "benefits of plant-derived substances relate to their phytochemical content which increase the quantity and diversity of gut microbiota".

As a consequence, I would expect data relative to microbiota taxa presence and abundance. I suggest inserting a table comparing taxa data where present and a brief discussion of their activity in connection with the pathology.

Author Response

The systematic review by Mackay and colleagues reports the effect of plant-derived substances in preventing necrotising enterocolitis. The review paper is well-architectured and the methodological approach based on prisma protocol is well thought out.

Ans:  Thanks for the encouraging comments.

As reported in the discussion section "benefits of plant-derived substances relate to their phytochemical content which increase the quantity and diversity of gut microbiota". As a consequence, I would expect data relative to microbiota taxa presence and abundance. I suggest inserting a table comparing taxa data where present and a brief discussion of their activity in connection with the pathology.

Ans: None of the included studies have reported information on microbiota taxa. We contacted the authors of the included studies to address this issue. The only 2 authors who responded, clarified that they did not collect such data.

Reviewer 2 Report

Comments and Suggestions for Authors

Systematic review of animal studies evaluating the effects of plant-derived substances on the prevention of NEC. This is an excellent well designed study on the subject. There some minor comments to the authors:

1.In figure 2 and 3, add  the overall  RR besides the one from each study.

2. Some minimal typos mistakes: line 124,  186, table 1 Yin (2020), 8th column: pyroptosis or apoptosis?

3. I would add as a limitation that almost all studies came from two countries: Turkey and China. I wonder why this subject does not awake interest in other parts of the world.

Author Response

Systematic review of animal studies evaluating the effects of plant-derived substances on the prevention of NEC. This is an excellent well-designed study on the subject. There are some minor comments to the authors:

1.In figure 2 and 3, add the overall RR besides the one from each study.

Ans: We had not pooled the data from the included studies considering the heterogeneity among the plant-derived substances. However, considering the possibility of shared mechanisms of benefits (e.g., anti-inflammatory effect, and protection from free radicle injury) of plant-derived substances as a class of intervention, we have included the overall relative risk (RR) in the Forest plots as suggested. We have modified Figure 2 and 3 and improvised the relevant sections of results, discussion (Highlighted), and the abstract. We have added a funnel plot for mortality (Figure 4) considering that data from 10 of the included studies that reported this outcome, has been pooled.

  1. Some minimal typos mistakes: line 124,186, table 1 Yin (2020), 8thcolumn: pyroptosis or apoptosis?

Ans: Thank you.

  • Line 124: We have added table 2 to the manuscript and removed e table 1.
  • Line186: Now we have modified.
  • Table 1 Yin: We provided the terminology as reported by the authors. They have specifically mentioned pyroptosis, which is distinct from apoptosis according to its definition.
  1. I would add as a limitation that almost all studies came from two countries: Turkey and China. I wonder why this subject does not awake interest in other parts of the world.

Ans: We agree that almost all included studies are from two countries. However, it is not the ‘limitation’ of our review. Discussing the possible reasons for this finding is beyond the scope of this manuscript. We believe that our review will serve the purpose and guide further research in this field.